# Validation and Optimization of PURE Ribosome Display for Screening Synthetic Nanobody Libraries

**DOI:** 10.3390/antib14020039

**Published:** 2025-05-02

**Authors:** Bingying Liu, Daiwen Yang

**Affiliations:** Department of Biological Sciences, National University of Singapore, 14 Science Drive 4, Singapore 117543, Singapore; liubingying@u.nus.edu

**Keywords:** synthetic nanobody library, PURE ribosome display, binder selection

## Abstract

Background/Objectives: PURE (Protein synthesis Using Recombinant Elements), an ideal system for ribosome display, has been successfully used for nanobody selection. However, its limitations in nanobody selection, especially for synthetic nanobody libraries, have not been clearly elucidated, thereby restricting its utilization. Methods: The PURE ribosome display selection process was closely monitored using RNA agarose gel electrophoresis to assess the presence of mRNA molecules in each fraction, including the flow-through, washing, and elution fractions. Additionally, a real-time validation method for monitoring each biopanning round was implemented, ensuring the successful enrichment of target protein-specific binders. The selection process was further optimized by introducing a target protein elution step prior to the EDTA-mediated disassembly, as well as by altering the immobilization surfaces. Finally, the efficiency of PURE ribosome display was enhanced by replacing the spacer gene. Results: The efficiency of PURE ribosome display was merely 4% with an unfavourable spacer gene. Using this spacer gene, EGFP- and human fatty acid-binding protein 4-specific nanobodies from a synthetic nanobody library were we successfully identified through optimizing the selection process. Choosing a spacer gene less prone to secondary structure formation increased significantly its efficiency in displaying synthetic nanobody libraries. Conclusions: Implementing a target protein elution step prior to EDTA-mediated disassembly and modifying the immobilization surfaces effectively increase selection efficiency. For PURE ribosome display, efficiency was further improved using a suitable spacer gene, enabling the display of large libraries.

## 1. Introduction

The growing demand for binders targeting specific proteins underscores the critical need for effective binder selection in protein engineering. Nanobodies, which are the antigen-binding fragments of heavy-chain-only antibodies, have proven highly valuable in research, diagnostics, and therapeutic applications [1,2,3]. Traditionally, functional nanobodies are sourced from immunized or naive libraries, but many laboratories face limitations due to the lack of access to the necessary animal facilities. Synthetic nanobody libraries present a viable alternative, being fully constructed from DNA oligonucleotides and thus avoiding the need for animal involvement. Nonetheless, synthetic nanobody libraries frequently contain numerous detrimental clones due to the lack of natural immune system evolution and errors that occur during library construction and screening processes. Consequently, a robust protein selection method is essential for identifying the rare favourable candidates.

To date, only a few synthetic nanobody libraries have been published [4,5,6,7,8,9,10,11,12]. While the methods for constructing and selecting antigen-specific nanobodies from synthetic nanobody libraries are well documented, their broad application is still under development. Phage display, which was established over 35 years ago, remains the most widely used method for protein selection [13]. However, its inherent limitation in library size constrains its application for very large synthetic libraries. In contrast, ribosome display uses ribosomal complexes formed during in vitro translation to link genotype with phenotype, thereby avoiding the use of living cells—the primary limiting factor for library size [14]. Although ribosome display based on *E. coli* S30 extracts has been developed over many years, its application to synthetic nanobody libraries remains limited. This is primarily due to reduced selection efficiency caused by the presence of RNases and proteases, which particularly hinders the recovery of low-copy or suboptimal members in synthetic nanobody libraries [15].

The reconstitution of the PURE system addresses this issue by eliminating RNases and proteases, resulting in a higher mRNA recovery rate compared to traditional S30 extracts [16,17]. However, the PURE system introduces new challenges due to its oversimplified nature. Comparative experiments have shown that, under identical conditions, the PURE system yields a greater quantity of protein than the S30 extract, likely due to the low protease content in the PURE system. However, enzyme activity assays have revealed higher enzyme activity in the S30 extract group than in the PURE system group, suggesting that a portion of the proteins produced in the PURE system are improperly folded, likely due to the absence of molecular chaperons [15]. Furthermore, Iwan Zimmermann and colleagues reported that three successive rounds of PURE ribosome display effectively identified maltose binding protein (MBP) binders from synthetic nanobody libraries. However, this approach proved inadequate for selecting nanobodies targeting membrane proteins [11,18]. The authors attributed this limitation to the inherent selection bias of ribosome display, which may favour certain background binders and impede the enrichment of target-specific nanobodies. Protein expression and presentation can vary across different selection systems; specific library members may fail to fold correctly in the PURE system yet manage to fold properly under intracellular expression conditions. Nevertheless, it is noteworthy that the enrichment of background binders should be largely influenced by the target protein display method and selection stringency, rather than the PURE ribosome display itself.

Addressing the limiting factors of the PURE system for screening synthetic nanobody libraries and subsequently optimizing the process can significantly enhance the utility of these libraries. In this study, we investigated the efficiency limitations imposed by mRNA secondary structure—an aspect overlooked during the transition from the S30 system to the PURE system. We employed a real-time analysis method to monitor the enrichment of specific binders, thereby significantly reducing troubleshooting time. Additionally, we developed strategies to facilitate the identification of nanobodies specific to target proteins by incorporating a target protein elution step prior to the EDTA-mediated disassembly, as well as altering the immobilization surface. Finally, we improved the PURE ribosome display system by altering the spacer genes to minimize mRNA secondary structure formation.

## 2. Results

### 2.1. Synthetic Nanobody Library Construction

The creation of a synthetic nanobody library involves the design of both the framework regions and complementarity-determining regions (CDRs). For the framework regions, the amino acid sequences were established based on the most frequently occurring sequences identified through the alignment of frameworks from ten previously published synthetic nanobody libraries (Figure 1). The framework designed here differs by only one amino acid from a previously published framework [9], which utilized a consensus framework sequence derived from llama genes—*IGHV1S1-IGHV1S1S5* (Figure 1).

The lengths of CDR1 and CDR2 were set at 10 amino acids, guided by the prevalent length observed in structurally characterized nanobodies. For CDR1, the first and last two amino acids were fixed as G, M, and G, based on their respective frequencies exceeding 30%. Similarly, in CDR2, the first two and last two amino acids were fixed as A, I, T, and Y, respectively. CDR3 was designed to be 12 amino acids long, while the first and last amino acids were fixed as A and Y, as their frequencies exceeded 50%. The remaining positions in the CDRs were fully randomized to introduce 19 amino acids at each position, with the exception of cysteine.

A standard nanobody was created to validate the universality of the framework design. The amino acid sequence of this representative nanobody was based on the consensus framework sequence derived from the library design, incorporating the most frequent amino acid in each CDR position (Appendix A). The standard nanobody was expressed in a soluble form and purified to about 85% purity using metal affinity chromatography with Ni-NTA agarose beads (Appendix A). Its melting temperature (Tm) was approximately 51 °C, which lies within the acceptable range of 50 to 70 °C for a nanobody (Appendix A).

The synthetic nanobody library was constructed by assembling the elements required for transcription and translation, including the T7 promoter, the nanobody pool, and a spacer gene derived from the M13 pIII high-GS-repeat region (Appendix A). To evaluate the library’s quality, a portion of the ligation products was transformed into *E. coli*, and 30 individual clones were randomly selected for sequencing. Among these, 20 contained full-length nanobodies with the correct reading frame (Figure 2). The incorrect clones were mainly due to single-nucleotide deletions, resulting in frameshifts in the sequences.

### 2.2. Target Protein Purification and Immobilization

The immobilization of target proteins is a crucial step for the selection of target-specific binders. Although passive adsorption is a widely employed technique for protein immobilization, it can lead to partial denaturation and result in non-uniform protein orientation. To overcome this, proteins are often fused with specific tags to preserve their native conformation and ensure consistent orientation. In this study, a streptavidin-binding peptide tag (SBP-tag) was utilized to immobilize the target protein on a streptavidin-coated surface. The SBP-tag is a 38-amino-acid peptide capable of binding to streptavidin with an equilibrium dissociation constant of 2.5 nM [21]. Using an SBP-tag to anchor target proteins onto streptavidin-coated solid surfaces not only preserves the native conformation of the target proteins but also allows biotin to be used for elution prior to the EDTA-mediated disassembly of ribosome–mRNA–protein (RMP) complexes. This method has proven effective in reducing non-specific binding, as will be discussed in the PURE ribosome display validation section.

EGFP was used as a model protein in the biopanning process; it is frequently employed due to its high solubility and distinct fluorescence characteristic [22,23,24]. In order to ensure complete exposure of the target protein surface, the SBP-tag was attached to the C-terminal, separated by a 20-amino-acid linker—(GGGGS)_4_—providing adequate spacing between the target protein and the SBP-tag (Appendix A). According to SDS-PAGE analysis, a significant portion of EGFP with the SBP-tag was successfully purified using immobilized metal affinity chromatography with Ni-NTA agarose beads (Appendix A). To confirm immobilization via the SBP-tag, purified EGFP-(GGGGS)_4_-SBP was incubated with streptavidin-coated magnetic beads. The SDS-PAGE results demonstrated that EGFP-(GGGGS)_4_-SBP remained bound to the streptavidin-coated magnetic beads even after three rounds of washing (Appendix A).

### 2.3. PURE Ribosome Display Validation

Although the PURE system has been proven to be a suitable system for ribosome display, its efficiency has not been clearly elucidated. To quantify the efficiency of the PURE ribosome display selection system, a well-characterized nanobody (Nb2) with high binding affinity to sfGFP (Kd = 15.8 nM) was employed [24]. The efficiency of the PURE ribosome display was assessed by quantifying the RMP complexes retained on an EGFP-coated surface. The PURE system contains the essential components for in vitro transcription and translation, enabling the use of either DNA or RNA as input material. In the assessment of ribosome display selection efficiency using DNA versus RNA, 200 ng of DNA or 2000 ng of mRNA was employed for translation. After a single wash, a comparison of the mRNA amounts retained in the elution fraction clearly showed that RNA as the translation resource resulted in higher efficiency than DNA (Figure 3A). This difference is likely due to the by-products of in vitro transcription, such as inorganic pyrophosphate, which may interfere with RMP complex formation. An analysis of the agarose gel revealed that the majority of mRNA was present in the flow-through fraction (Figure 3A), suggesting a lack of formation of functional RMP complexes.

To investigate the underlying cause of deficiency in functional RMP complex formation, the transcribed proteins were analyzed by Western blot using anti-FLAG antibodies. The result showed that full-length products comprised only a small fraction of the total (Figure 3B). Predominantly, two bands of around 17 kDa were observed, indicating premature termination or pausing within the middle region of the mRNA. This region corresponds to the 5′ half of the spacer gene, which has a high GC content (~70%) and is predicted to form stable RNA secondary structures (using RNAfolder web server) [25]. These secondary structures likely hinder ribosome progression, as the simplified PURE system lacks certain enzymes such as RNA helicases to unwind such structures. Various protein fragments have been utilized as spacer genes in ribosome display with cell extracts, including the glycine- and serine-rich region of M13 phage g3p [26], the Cκ domain of a light chain of an antibody [27], a helical segment of *E. coli* TolA [28], and the λ phage protein D [29]. However, few studies have explored whether these spacer genes are suitable for PURE ribosome display. Because ribosome movement is hindered by mRNA secondary structures, extending the in vitro translation incubation time can increase the likelihood of ribosomes reaching the end of the mRNA. The following two incubation times were compared: a 5 min period commonly used for ribosome display with S30 crude extract, and a longer 30 min period as reported by Iwan Zimmermann and colleagues [11]. The agarose gel comparison showed that the 30 min incubation resulted in slightly higher amounts of functional RMP complexes, supporting the hypothesis that longer incubation times can mitigate the effects of mRNA secondary structures (Figure 3A).

With approximately 10^13^ input mRNA molecules (roughly equivalent to the number of ribosomes), the amount of mRNA retained in the elution fraction after two washes was approximately 4 × 10^11^, as measured by RT-qPCR. This indicates that the efficiency of PURE ribosome display using the M13 pIII spacer gene is about 4%, a figure comparable to the reported efficiency of S30 extract ribosome display for a short His6-tag peptide, which is approximately 5% [30]. To ensure the efficient display of each library member, about 10^13^ input mRNA molecules are required when the diversity of the synthetic nanobody library is around 10^11^ at the DNA level, thereby allowing each member to form about four functional RMP complexes in the translation process.

### 2.4. Real-Time Validation of the Enrichment

Given that the efficiency of PURE ribosome display was suboptimal, an effective monitoring system is essential for successfully selecting nanobodies specific to the target proteins. The selection process typically involves multiple rounds to ensure the dominance of target protein-specific nanobodies. Assessing the successful enrichment of target protein binders has long been a challenge for researchers. Many testing methods rely on comparing the retention of molecules between groups coated with the antigen and the negative control group in the later stages of biopanning [11]. However, this approach can lead to time-consuming troubleshooting if issues arise in the early biopanning rounds. In our work, we applied a real-time validation method for the enrichment process. This method enables monitoring of each biopanning round to verify the successful enrichment of target protein-specific binders, as described below.

mRNA encoding Nb2 was introduced into the synthetic nanobody library at an Nb2/nanobody ratio of 10^2^:10^13^ to monitor the enrichment of an Nb2 clone. As mentioned above, the ratio of functional RMP complexes to input mRNA molecules was approximately 4%. With 100 mRNA molecules for the Nb2 clone in the input, about four functional RMP complexes containing Nb2 were present in the first round of panning, ensuring identification of this nanobody in subsequent processes. The diversity of our nanobody library was estimated to be around 2 × 10^11^, and the mRNA production step amplified each nanobody sequence approximately 50-fold by T7 RNA polymerase, ensuring the successful formation of functional RMP complexes for each sequence. A specialized primer was designed to specifically bind to the Nb2 sequence rather than other nanobodies, facilitating the assessment of the presence and proportion of the Nb2 clone within a pool of nanobody clones (Figure 4A).

An analysis of the agarose gel image, which assessed the proportion of the Nb2 clone among the RT-PCR products of the total nanobodies (Figure 4B(a)), clearly demonstrated a gradual increase in the Nb2 proportion throughout the biopanning process. Following the first round of biopanning, Nb2 in the RT-PCR products could only be detected after 25 PCR cycles. However, after the fourth round, Nb2 in the RT-PCR products could be detected after only 10 PCR cycles (Figure 4B(a)). Approximately 0.1 ng of RT-PCR products from the fourth round was used as PCR templates. The amount of Nb2 after 10 PCR cycles was quantified to be approximately 10 ng, as measured by the agarose gel electrophoresis image. Therefore, the initial amount of Nb2 in the 0.1 ng RT-PCR product was estimated to be around 0.01 ng (10ng/2^10^), indicating that Nbs accounted for roughly 10% of the total nanobodies after four rounds of panning. To further accelerate the dominance of target-specific nanobodies, a protein elution step was developed prior to the conventional EDTA-mediated disassembly step to decrease non-specific binder release. With the optimization of the elution step (Appendix A), the Nb2 sequence was detected after only 20 cycles following the first round of biopanning, which is 5 cycles fewer than with the conventional EDTA elution method (Figure 4B(b)).

Utilizing this optimized method (Appendix A), a single-clone ELISA analysis was conducted to identify the nanobodies targeting EGFP after the fourth round of biopanning. About 10 of the 87 colonies evaluated showed an A450 ratio (EGFP-coated well/BSA-coated well) exceeding 1.9, suggesting the potential presence of EGFP binders (Figure 4C). Sequencing results revealed that 9 out of these 10 clones shared the same nanobody sequence, designated as H1, while the remaining 1 was labeled as D4. To further characterize the interaction between H1/D4 and EGFP, pull-down assay, size exclusion chromatography (SEC), and isothermal titration calorimetry (ITC) were employed.

A pull-down assay was conducted by employing His-tagged nanobodies as bait to capture EGFP in solution. An analysis of the SDS-PAGE revealed that, after stringent washing, a significant quantity of EGFP, EGFP-(GGGGS)_4_-SBP and its degraded products were captured by either H1 or D4. D4 did not bind to EGFP; instead, it bound to the C-terminal (GGGGS)_4_-SBP fusion part, as evidenced by the lack of EGFP in the elution sample (Figure 5A). Therefore, the following characterizations were focused on the interaction between H1 and EGFP.

H1 exhibited an SEC (Superdex 75 10/300 GL column) profile featuring a single symmetrical peak at the anticipated volume (13.7 mL), indicating a uniform species. Subsequently, a mixture of EGFP and H1 was introduced into the column to assess potential co-elution. An analysis of the SEC profile revealed the evident co-elution of EGFP and H1, confirming their interaction (Figure 5B). The binding affinity was quantified using ITC. The binding affinity between H1 and EGFP was determined to be 116 ± 21.3 nM (Appendix A).

### 2.5. Biopanning the Synthetic Nanobody Library Against hFABP4 by Altering the Surface for Immobilization

To further test the robustness of the above-mentioned selection process, another target protein—human fatty acid binding protein 4 (hFABP4)—was used. FABPs transport cargos such as long-chain fatty acids, eicosanoids, and other lipids inside cells, facilitating their transport to designated compartments [31]. In addition to their intracellular roles, some FABPs are also found in blood circulation. Substantial evidence indicates that serum FABP4 levels are significantly elevated in patients with metabolic disorders such as diabetes, insulin resistance, obesity, and atherosclerosis [32]. Structural analyses have revealed that local unfolding of the second helix creates a gate for ligands to enter the binding pocket [33]. Based on the molecular mechanism underlying fatty acid entry and exit, inhibiting the conformational change in the helical cap of hFABP4 appears to be an effective strategy for inhibiting its function. A key application of nanobodies in research involves stabilizing specific conformational states of dynamic proteins, such as GPCRs [34,35]. Furthermore, no nanobody targeting hFABP4 has been identified to date.

Using the biopanning method described above, no hFABP4-specific nanobodies emerged after four rounds of biopanning. Instead, most clones obtained after the 4th round contained a streptavidin-binding motif—HPQ/M—in their CDRs, indicating an enrichment of streptavidin-specific binders [36]. The HPQ/M motif was bound to streptavidin in the biotin binding site and could be released from streptavidin under biotin elution conditions. The issue of enriching streptavidin-specific binders was not observed during the EGFP biopanning process but emerged as a significant problem during the hFABP4 biopanning process. The major reason might be the smaller size of hFABP4 (approximately 14 kDa) compared to EGFP (approximately 27 kDa). Consequently, under identical coating conditions, hFABP4 occupied less space than EGFP, allowing RMPs more opportunities to interact with streptavidin or the solid surface.

To reduce the enrichment of streptavidin-specific binders, a pre-incubation step was introduced after the first round of biopanning. In this step, the ribosome display system was first incubated with a streptavidin-coated surface to remove streptavidin-specific RMPs. Subsequently, the flow-thorough was collected and incubated with an hFABP4-coated surface. Additionally, an alternative surface (Ni-NTA agarose beads) for immobilizing hFABP4 was applied in the third round, avoiding exclusive reliance on streptavidin-coated solid surfaces (Appendix A). After the fourth round of biopanning, the sequences in the nanobody pool were cloned into protein expression vectors for subsequent single-clone ELISA analysis. Out of the 95 individual colonies tested, 4 colonies exhibited an A450 ratio (hFABP4-coated well/BSA-coated well) exceeding 1.9, indicating potential binding to hFABP4. Among these four clones, two of them demonstrated binding to hFABP4 in the pull-down assay, identified as F4_B6 and F4_C9. To evaluate the specificity of the identified hFABP4-specific nanobodies, the hFABP3 protein—sharing 63.9% sequence identity and 75.9% similarity with hFABP4—was utilized as a control. Neither of these two nanobodies exhibited binding to hFABP3 (Figure 6A).

F4_C9 has a mutation in the framework region 3 (CAG to CAT, resulting in Q to H), which possibly happened during the RT-PCR process due to the use of a non-proofreading Taq polymerase. After mutating H back to Q, F4_C9 showed strong interaction with sugar-based polymer, similar to clone D4. For F4_B6, its expression yield and solubility were optimal, facilitating a straightforward purification process using immobilized metal affinity chromatography with Ni-NTA agarose beads. The elution fraction was subsequently subjected to SEC using a Superdex 75 10/300 GL column. The SEC profile of F4_B6 displayed a single symmetrical peak at the expected elution volume (13.0 mL), along with an additional smaller peak in front of it, likely indicative of either dimer formation due to inter-molecular disulfide bonds or dimerized nanobodies, confirmed by analyzing non-reducing protein samples on SDS-PAGE (Figure 6B). This implied that a small amount of F4_B6 might not have been properly folded, leading to the exposure of cysteine residues intended for intramolecular disulfide bond formation. Fractions from the major symmetrical peak were collected for the binding affinity measurement.

According to the SEC profile of the hFABP4 and F4_B6 mixture, the elution volume of the major peak shifted to 11.4 mL. The following two additional small peaks were observed: one at 10.5 mL, attributed to a small amount of hFABP4 dimer and F4_B6 dimer, and another at 12.9 mL, corresponding to residual hFABP4 without interaction with F4_B6. The co-elution of hFABP4 with F4_B6 further affirmed their interaction (Figure 6B). The binding affinity between them was measured by ITC and found to be approximately 295 nM ± 91.7 nM (Figure 6C).

### 2.6. PURE Ribosome Display Optimization by Altering Spacer Gene

The spacer derived from natural M13 pIII is Gly- and Ser-rich in the N-terminal region. The corresponding RNA sequence of this region, about 150 nucleotides long, has an average GC content of ~70%, which was predicted to have a high likelihood of forming stable RNA secondary structure. This stable secondary structure might serve as a protection mechanism against intracellular RNase degradation. Another commonly used spacer gene, *E. coli* TolA, consisting of multiple Ala-Lys-Glu repeats with an average GC content of ~56%, was predicted to be significantly less prone to forming RNA secondary structure than the spacer gene from M13 pIII. To further confirm the effect of spacer genes for PURE ribosome display efficiency, Nb2 was cloned into vector pRDV. After incubation for 30 min, it was clearly shown that the majority of the translated proteins were full-length products, indicating that most ribosomes were able to reach the end of the mRNA (Figure 7A). Since no factors impeded the speed of translation, 30 min might be a long incubation time for the in vitro translation of an mRNA molecule with 800 nucleotides, which increases the possibility of ribosome release. By comparing the RMP complexes retained in the elution fraction between samples with 5 min and 30 min incubation times, it is clear that significantly more RMP complexes were retained in the 5 min sample than the 30 min sample (Figure 7B). For the 5 min incubation sample, the amount of the 0.8 kb mRNA retained in the elution fraction after two washes was approximately 20% of the input amount, as measured by RT-qPCR. This suggests that the efficiency of PURE ribosome display using the TolA spacer gene is about 20%.

## 3. Discussion

The application of PURE ribosome display has been somewhat limited by the lack of a comprehensive understanding. Although most principles are the same for ribosome display, whether using cell extract or the PURE system, some factors have been undervalued in terms of their impact on selection efficiency. In this study, we highlighted the importance of spacer gene choice, which has often been overlooked when applying designs from ribosome display using cell extracts. The incubation time for the in vitro translation process also needs to be fine-tuned for different spacer genes, as ribosome movement speed varies across sequences. An incubation time that is too short might result in immature products, while one that is too long could cause ribosome complexes to dissociate from the mRNA.

In this work, a synthetic nanobody library was constructed by employing the amino acids with the highest frequencies, derived from a multiple sequence alignment of previously published synthetic nanobody libraries, as the framework sequence. This framework was combined with full randomization of three CDRs. Trimer phosphoramidites were utilized during the synthesis of degenerate primers to control amino acid types and minimize frameshifts, like single-base deletions, which may occur in the CDRs during primer synthesis. Furthermore, this method prevented the inclusion of rare codons that could affect protein yield, potentially leading to expression level biases during culturing in 96-well plates for the single-clone ELISA analysis. The incorporation of Nb2 into the synthetic nanobody library enables real-time confirmation of the successful enrichment of EGFP-specific binders through a straightforward PCR test. This method can serve as a preliminary step for verifying the functionality of the entire system and is particularly beneficial for beginners using PURE ribosome display or when utilizing new libraries.

Non-specific binding consistently presents a challenge by hindering the effective enrichment of specific binders, regardless of the selection methods employed. To reduce non-specific binding, several strategies are commonly employed. One approach involves a pre-incubation step, where the display system is exposed to the immobilized surface without the antigen, aiming to eliminate surface-specific binders. Another strategy involves altering the chemistry of the immobilized surface, such as utilizing magnetic beads, Ni-NTA agarose beads, and 96-well plates, as employed in this study. Alternatively, altering the protein on solid surfaces itself can be effective, for instance, by utilizing streptavidin- or avidin-coated solid surfaces as substitutes. Additionally, adjusting selection pressure factors like washing stringency or gradually reducing the concentration of the target protein can also reduce non-specific binding. Approaches such as prolonging washing incubation time, increasing the number of washing steps, or adjusting detergent concentration have been shown to effectively enhance the washing efficiency for removing non-specific binding.

Through the thorough examination of ribosome display using the PURE system, a clearer understanding of its strengths and limitations has emerged. The robustness of PURE ribosome display was validated using our in-house synthetic nanobody library, successfully screening for EGFP- and hFABP4-specific nanobodies, even with the use of an unpreferred spacer gene, M13 pIII. Following this validation and optimization, the application of PURE ribosome display for screening large libraries is expected to grow in the near future.

## 4. Conclusions

In this work, we highlighted the importance of spacer gene choice in ribosome display when using the PURE system. To better evaluate and troubleshoot the lengthy biopanning process, we applied the direct RNA agarose gel analysis method and a real-time analysis method. The incorporation of a protein elution step before the EDTA-mediated disassembly and the alteration of immobilization surfaces significantly increased selection efficiency. This optimization enabled the successful selection of EGFP- and hFABP4-specific nanobodies from a synthetic nanobody library. Our findings provide a more comprehensive understanding of PURE ribosome display, facilitating its future application in binder selection.

## 5. Materials and Methods

### 5.1. Protein Expression and Purification

Plasmids containing His-EGFP-(GGGGS)_4_-SBP or His-SUMO-hFABP4-(GGGGS)_4_-SBP were transformed into *E. coli* BL21 (DE3) cells for the overexpression of recombinant proteins. *E. coli* BL21 cells containing the desired plasmid were grown in 1 litre of LB broth (containing 0.1 mg/mL ampicillin or 0.05 mg/mL kanamycin) to an OD600 of 0.6 at 37 °C. Subsequently, the temperature was lowered to 20 °C, and the cells were induced with 200 µM IPTG for 18 h. Harvested cells were disrupted using a sonicator at 30% sonication amplitude, with 1 s on/off intervals, in PBS buffer (137 mM NaCl, 2.7 mM KCl, 10 mM Na_2_HPO_4_, 1.8 mM KH_2_PO_4_, pH 7.4). The lysate resulting from sonication was subjected to centrifugation at 8000× *g* and 4 °C for 40 min to separate the soluble and insoluble components. The supernatant, containing the soluble fraction, was loaded onto a column containing 5 mL Ni-NTA beads (QIAGEN, Singapore, Singapore). The slurry was incubated at 25 °C for 1 h on a rocker at 30 rpm. After washing out the impurities, the proteins were eluted with 10 mL PBS containing 200 mM imidazole. Following cleavage of the His-tag or SUMO-tag by thrombin or ULP1, the proteins were dialyzed against PBS to remove imidazole and His-tag. For hFABP4, the protein solution was reloaded onto a Ni-NTA column to remove the SUMO-tag. The proteins were concentrated using centrifugal filters with a 3 kDa cut-off and separated by size exclusion chromatography using a Superdex 75 10/300 GL column (Cytiva, Singapore, Singapore) in PBS.

### 5.2. Circular Dichroism Spectroscopy

Circular dichroism (CD) spectroscopy was employed to assess the thermal stability of the nanobodies. The protein sample, maintained at a concentration of 0.2 mg/mL, was placed in a phosphate buffer (20 mM, pH 7.1). CD spectra were recorded using a JASCO J-1100 CD Spectrophotometer (Tokyo, Japan) equipped with a thermal controller. For the thermal denaturation experiment, the temperature was incrementally increased from 20 °C to 90 °C at a rate of 2 °C/min. The change in ellipticity at 219 nm with respect to temperature was employed to determine the melting temperature (Tm) of the nanobodies.

### 5.3. RNA Bleach Agarose Gel

To protect RNA from degradation by RNase, a bleach agarose gel was prepared to minimize RNase activity. This involved adding 0.5 g of agarose into 50 mL of TBE buffer (89 mM Tris, 89 mM boric acid, 2 mM EDTA, pH ~8.3) containing 0.6% sodium hypochlorite. After incubation at room temperature for 5 min, the agarose was dissolved by microwaving the solution for 2 min. SYBR^TM^ Green II RNA gel stain was then added at a dilution of 1:10,000. The resulting gel solution was poured into a gel tank for solidification. Electrophoresis was carried out at 140 V for 20 min on ice.

### 5.4. RT-qPCR

The RT-qPCR analysis was carried out utilizing the NEB Luna^®^ Universal One-Step RT-qPCR Kit. Each reaction was conducted in a 20 µL volume, comprising 10 µL of Luna Universal One-Step Reaction Mix, 0.4 µM of each primer (RT_universal_Fr and RT_universal_Rv), 1 µL of 20 × Luna WarmStart^®^ RT Enzyme Mix, and 2 µL of 10 × diluted mRNA. The cycling conditions involved a 10 min incubation at 55 °C for reverse transcription, followed by initial denaturation at 95 °C for 1 min. Subsequently, 40 cycles were performed with denaturation at 95 °C for 10 s and annealing at 60 °C for 30 s. Reactions were run on a Bioer LineGene 9600 Plus machine.

To generate a standard curve, a dilution series of mRNA corresponding to a specific nanobody was used. The initial concentration was determined using NanoDrop UV–Visible Spectroscopy. The quantity of eluted mRNA was then determined by referencing the standard curve.

### 5.5. Synthetic Nanobody Library Construction

Primers containing randomized regions were created using trimer phosphoramidites (GenScript, Singapore, Singapore). The full-length nanobody nucleotide sequence was divided into seven segments denoted as FR1, CDR1, FR2, CDR2, FR3, CDR3, and FR4. Except for FR3, which was synthesized through a standard PCR procedure, the other six segments were obtained as primers (FR1_Fr, CDR1_Rv, FR2_Fr, CDR2_Rv, CDR3_Fr, FR4_Rv) (Appendix A). These primers were assembled and amplified via overlap extension PCR using Q5 DNA Polymerase (NEB), as depicted in Appendix A. Each 50 μL overlap extension PCR reaction contained 10 μL of 5 × Q5 reaction buffer, 0.4 mM dNTPs, 0.4 μM forward primer, 0.4 μM reverse primer, and 0.5 μL of Q5 DNA polymerase. The annealing temperatures for the three PCR experiments were 67 °C (CDR3), 67 °C (CDR1), and 60 °C (CDR2). Each reaction consisted of 30 cycles with an extension time of 3 s per cycle. The 3 resulting double-stranded PCR products and FR3 were assembled through another round of overlap extension PCR. This second round involved two sets of PCR conditions to fuse the DNA fragments and amplify the product. To assemble FR3 and CDR3-FR4, the reaction consisted of an initial 5 cycles at 61 °C with a 2 s extension time, followed by 20 cycles at 57 °C with a 3 s extension time. To assemble FR1-CDR1 and FR2-CDR2, the procedure included an initial 5 cycles at 70 °C with a 2 s extension time, followed by 20 cycles at 59 °C with a 3 s extension time. The products obtained from these reactions were then fused using the same overlap extension PCR method (Appendix A). As the melting temperatures of both the overlap regions and the primers were consistent at 67 °C, the final PCR reaction was streamlined to 20 cycles at 67 °C.

The complete nanobody constructs were subjected to double digestion with BamHI and HindIII, then ligated into similarly double-digested ribosome-display vectors using T4 DNA ligase (NEB). The 300 μL ligation mixture contained 30 μL of 10× T4 ligation buffer, 10 μL T4 DNA ligase, 25 μg of double-digested vector, and 10 μg of double-digested insert. The ligation reaction was carried out at 16 °C for 16 h, after which the T4 ligase was inactivated at 65 °C for 15 min. The ligation mixture was purified using the NEB Monarch^®^ PCR Cleanup Kit. The purified ligation product served as the template for generating the transcription template. The forward primer (T7_transcript_Fr) was designed to anneal upstream of the T7 promoter, and the reverse primer (T7_transcript_Rv) annealed at the 3′ stem loop region. PCR was carried out in a 50 μL reaction comprising 10 μL of 5× Q5 reaction buffer, 0.4 mM dNTPs, 0.4 μM forward primer, 0.4 μM reverse primers, 100 ng of the ligation product, and 0.5 μL Q5 DNA polymerase. The cycling conditions included 20 cycles with an annealing temperature of 70 °C. The PCR products were verified by agarose gel electrophoresis and purified using the QIAGEN gel purification kit.

In vitro transcription reaction was carried out utilizing the RiboMAXTM Large-Scale RNA Production System (T7) from Promega. A 50 μL reaction comprised 10 μL of 5× T7 transcription buffer, 15 μL of rNTPs (25 mM each), 5 μL of Enzyme mix, and 0.8 μg of DNA templates. The reaction was incubated at 37 °C for 2 h. Subsequently, 1 μL of DNase I was added to digest the DNA templates at 37 °C for 15 min. The mRNA was purified using the NEB Monarch^®^ RNA Cleanup Kit, flash-frozen in liquid nitrogen, and stored at −80 °C.

### 5.6. PURE Ribosome Display Screening

The PUREfrex2.1 Cell-Free Protein Synthesis Kit (GeneFrontier Corporation, Kashiwa, Japan) was used to display synthetic nanobody libraries on ribosomes. Each 10 μL reaction contained 4 μL of Solution I (amino acids, NTPs, tRNAs and substrates for enzymes), 0.5 μL of Solution II (proteins in 30% glycerol buffer), 1 μL of Solution III (20 µM ribosome), 0.5 μL of 10 mM cysteine, 0.5 μL of 80 mM reduced glutathione (GSH), 0.5 μL of 60 mM oxidized glutathione (GSSG), 0.5 μL of 1.875 mg/mL DsbC (disulfide bond isomerase), and 2.5 μL of mRNA solution (approximately 2 μg/μL). After incubation at 37 °C for 30 min, 100 μL of ice-cold biopanning buffer (WTB-BSA: 50 mM Tris-acetate pH 7.4, 150 mM NaCl, and 50 mM magnesium acetate, 0.5% BSA) supplemented with 2.5 μL of 0.2 g/mL heparin solution and 1 μL of SURPEase-In^TM^ RNase inhibitor was added to the ribosome display system and maintained at 4 °C for the subsequent biopanning process. The ribosome display mixture was applied to antigen-coated surfaces and incubated at 4 °C for 1 h with shaking at 30 rpm on a rocker.

Following incubation, unbound or weakly bound components were removed by washing with WTB or WTB-T (50 mM Tris-acetate pH 7.4, 150 mM NaCl, and 50 mM magnesium acetate, 0.1% Tween-20). The washing stringency varied across rounds of panning (Appendix A). For bead-based washes, 500 µL of WTB was utilized per wash with 3 pipetting cycles, followed by a 5 min incubation to enable magnetic beads separation. For 96-well plate washes, 300 µL of WTB was used with 3 pipetting cycles and no incubation.

The retained RMP complexes were disassembled using 100 μL of elution buffer (50 mM Tris-acetate pH 7.4, 150 mM NaCl, 50 mM EDTA, 100 µg/mL yeast RNA), and the resulting mRNA was purified using the NEB Monarch^®^ RNA Cleanup Kit. Following optimization, proteins were disassociated from streptavidin by biotin prior to RMP disassembly by EDTA. The purified mRNA was subject to reverse transcription and amplification using the QIAGEN OneStep RT-PCR Kit to produce DNA templates for the next round of selection. Each 50 μL RT-PCR reaction included 10 μL of 5X QIAGEN OneStep RT-PCR buffer, 0.4 mM dNTP mix, 0.6 μM of each primer (RT_universal_Fr, RT_universal_Rv), 2 μL of QIAGEN OneStep RT-PCR Enzyme mix, and 7 μL of eluted mRNA (equivalent to half of the total amount). Reverse transcription was performed at 50 °C for 30 min. PCR amplification followed an annealing temperature of 50 °C for 25 cycles. The resulting PCR products were purified, digested with BamHI and HindIII, and ligated into the ribosome display vector using T4 DNA ligase. Ligation was carried out at 16 °C for 16 h. The purified ligation product was then utilized as the template for PCR to generate the transcription template for the next selection round.

### 5.7. Single-Clone ELISA Analysis of Potential Binders

Single-clone ELISA analysis was conducted using a MaxiSorp 96-well plate. Each well was coated with 100 µL of 0.01 mg/mL target protein or BSA (used as negative control) in the ELISA coating buffer (50 mM carbonate–bicarbonate buffer, pH 9.6), followed by incubation at 4 °C for 16 h. The plate was then washed three times with 200 µL of PBS and blocked with 200 µL of blocking buffer (2% skimmed milk in PBS) at 25 °C for 1 h. After discarding the initial blocking buffer, 50 µL of fresh blocking buffer was added to each well, and the plate was ready for use.

The RT-PCR products from the final round were ligated into a protein expression vector and transformed into *E. coli* BL21 cells. Individual clones were cultured and induced in a 96-well deep-well plate (containing 1 mL of medium per well). Protein expression was induced with 200 µM IPTG at 20 °C for 16 h. Following induction, cells were harvested by centrifugation at 500× *g* for 10 min and then resuspended in 200 µL of resuspension buffer (PBS supplemented with 2 mM EDTA, and 0.1% Triton X-100). The cell resuspension underwent one freeze–thaw cycle, with incubation at −80 °C for 10 min followed by 37 °C for 10 min. Subsequently, 20 µL of freshly prepared lysozyme solution (10 mg/mL) was added to each well, and the plate was incubated at 37 °C for 30 min. A second freeze–thaw cycle was performed to further lyse the *E. coli* cells. Following this, 0.02 units of DNase I in 100 mM MgCl_2_ solution was added to each well, and the plate was incubated at 25 °C for 15 min. Finally, the plate was centrifuged at 3500× *g* for 30 min to separate the soluble and insoluble fractions.

The supernatant, containing nanobodies tagged with 6-histidines, was transferred to the wells coated with either target proteins or BSA. After 1 h of incubation, the wells were washed three times with 200 µL of PBS-T (0.1% Tween-20). Next, 100 µL of a 1:10000 dilution HisProbe^TM^-HRP (Thermo Scientific, Singapore, Singapore) in PBS was added to each well and incubated for 1 h. Excess probe was removed by three additional washes with 200 µL of PBS-T (0.1% Tween-20). Then, 100 µL of the 1-step TMB ELISA substrate solution (3,3′,5,5′-Tetramethylbenzidine, Thermo Scientific) was added to each well and incubated at room temperature for 15 min. Finally, the chromogenic reaction was stopped by adding 2 M sulfuric acid, and the absorbance was measured at 450 nm.

### 5.8. Pull-Down Assay

Candidate binders identified via the single-clone ELISA experiments were upscaled in a 10 mL cell culture for further binding analysis. The resulting cell pellet was resuspended in 1 mL of PBS and subjected to sonication for 60 s (25% sonication amplitude with 1 s on/off intervals). A 300 µL aliquot of the resulting supernatant was mixed with 20 µL of Ni-NTA agarose resin to selectively capture His-tagged nanobodies. Unbound proteins or impurities were removed through four washes with 500 µL of PBS (with 10 mM imidazole included in the 3rd and 4th washes). Subsequently, 100 µL of target protein solution (approximately 0.2 mg/mL) was added to the nanobody-coated beads. After three additional washes with 500 µL of PBS (with 10 mM imidazole in the 3rd wash) to eliminate non-specific binding, bound proteins were eluted using 50 µL of PBS containing 200 mM imidazole. The eluted fractions were then analyzed by SDS-PAGE to assess protein composition.

### 5.9. Isothermal Titration Calorimetry

The binding affinities between the identified nanobodies and their respective target proteins were evaluated using a MircoCal PEAQ ITC instrument at 25 °C. After gel filtration chromatography, both nanobodies and target proteins were exchanged into the same buffer (PBS). Protein concentrations were determined using a NanoDrop UV–Visible Spectrophotometer. Target proteins were loaded into the cell holder at a concentration of 10 µM, while the nanobodies were loaded into the syringe at 150 µM. The experimental protocol included an initial injection of 0.4 µL, followed by 19 injections of 2 µL each, with 150 s intervals between the injections. The stirring speed was set at 500 rpm. Subsequently, the raw data were processed and then fitted to a one binding site model.

## Figures and Tables

**Figure 1 antibodies-14-00039-f001:**
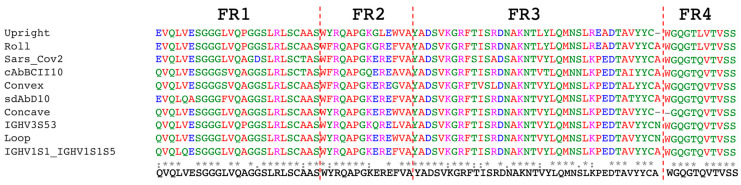
Multiple sequence alignment of framework regions from 10 published synthetic nanobody libraries, and the one (sequence on the bottom) designed in this study [8,9,10,11,12,19,20]. *, :, and blank represent fully conserved, partially conserved, and non-conserved residues across all sequences at a given position, respectively.

**Figure 2 antibodies-14-00039-f002:**
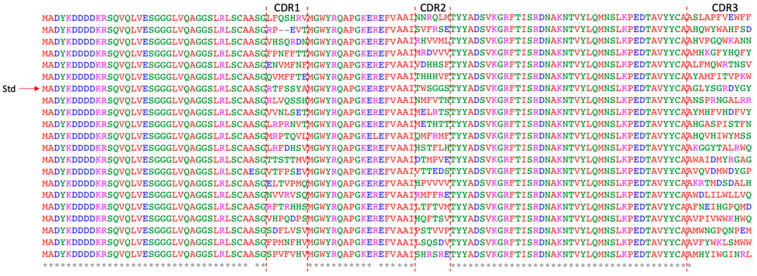
Multiple sequence alignment of 20 in-frame nanobody sequences with the standard nanobody sequence. ‘Std’ stands for the standard nanobody sequence, which was used as a control to compare with other sequences. * and blank represent fully conserved and non-conserved residues across all sequences at a given position, respectively.

**Figure 3 antibodies-14-00039-f003:**
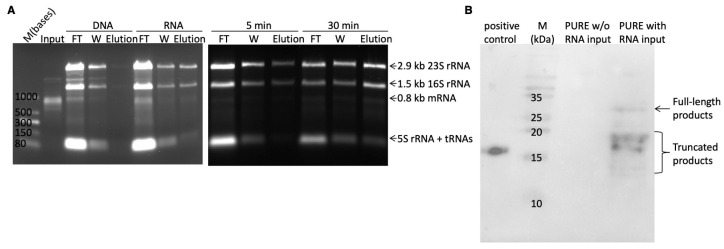
PURE ribosome display validation and optimization. (**A**) RNA agarose gel image for the efficiency test of PURE ribosome display. Left panel: Use of DNA versus RNA as input materials for the PURE system. Right panel: 5 min versus 30 min incubation times. The RNA size for nanobody together with spacer gene was 789 bases. ‘FT’ and ‘W’ represent flow-through and wash fraction. (**B**) Western blot image of proteins synthesized by PURE system. The positive control was the standard nanobody with FLAG-tag on its N-terminal and no spacer. The full-length proteins consist of 263 amino acids (~27 kDa).

**Figure 4 antibodies-14-00039-f004:**
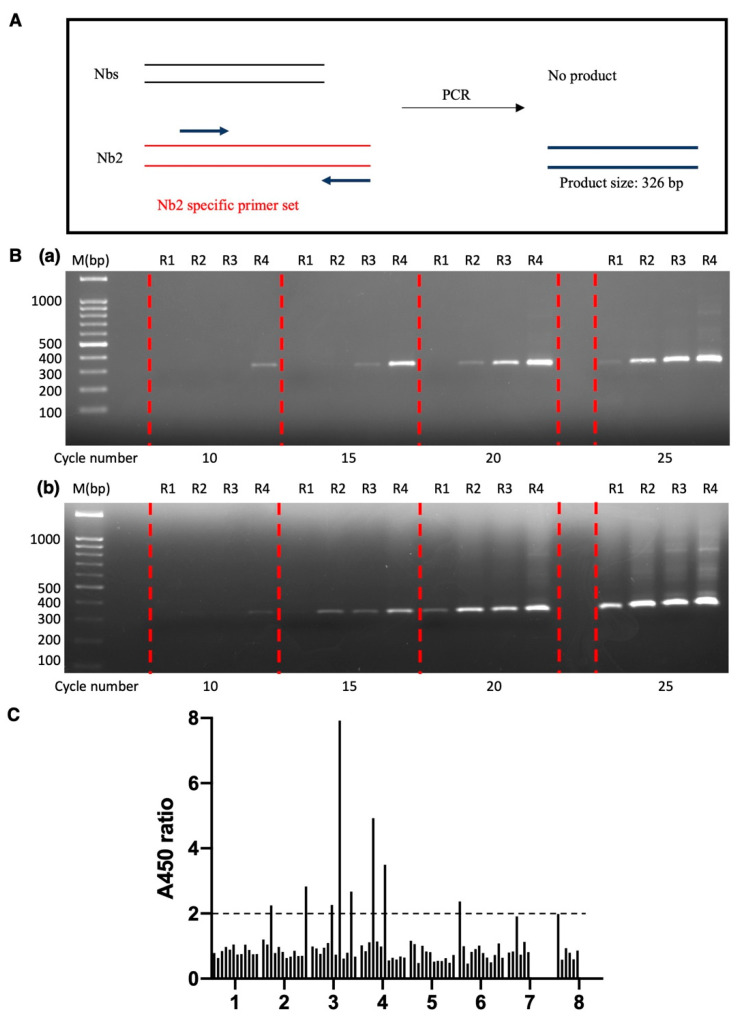
Real-time validation of EGFP-specific binder enrichment. (**A**) Diagram of PCR for testing the presence and proportion of the Nb2 clone in the nanobody library after every selection round. (**B**) Agarose gel electrophoresis image of the PCR products corresponding to the Nb2 clone. R1: 1st round; R2: 2nd round; R3: 3rd round; R4: 4th round. (**a**) EDTA elution method. (**b**) Biotin elution method. (**C**) Single-clone ELISA result after the 4th biopanning round using biotin elution method. A450 ratio: A450 from EGFP-coated well/A450 from BSA-coated well.

**Figure 5 antibodies-14-00039-f005:**
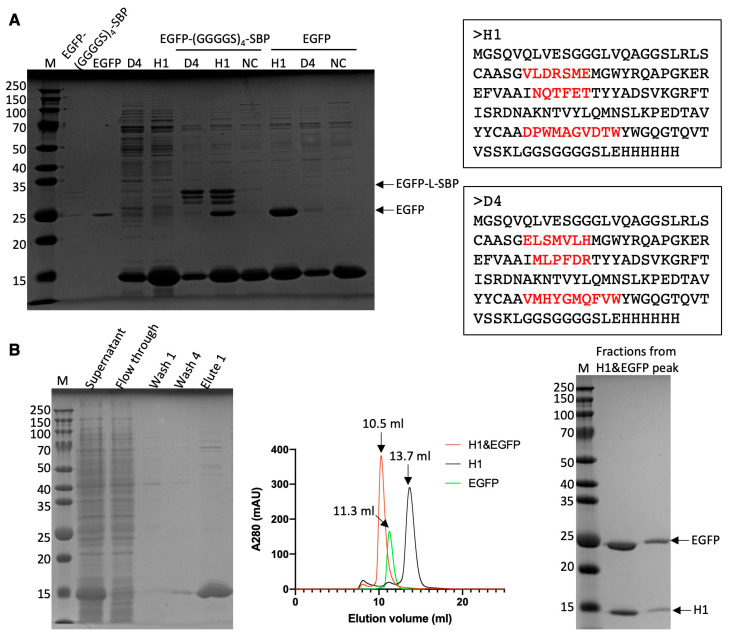
Analysis of the interaction between EGFP and EGFP-specific nanobodies—H1 and D4. (**A**) SDS-PAGE image of the pull-down assay, including the Ni-NTA beads coated by nanobodies, the elution fraction after incubation with EGFP/EGFP-(GGGGS)4-SBP. NC refers to the negative control, which is a nanobody that does not interact with EGFP. (**B**) SDS-PAGE image of the purification process of H1 (left panel), SEC profile (middle), and SDS-PAGE of the peak fraction for the mixture of H1 and EGFP. Wash 1: 0 mM imidazole; Wash 4: 20 mM imidazole; Elute 1: 200 mM imidazole.

**Figure 6 antibodies-14-00039-f006:**
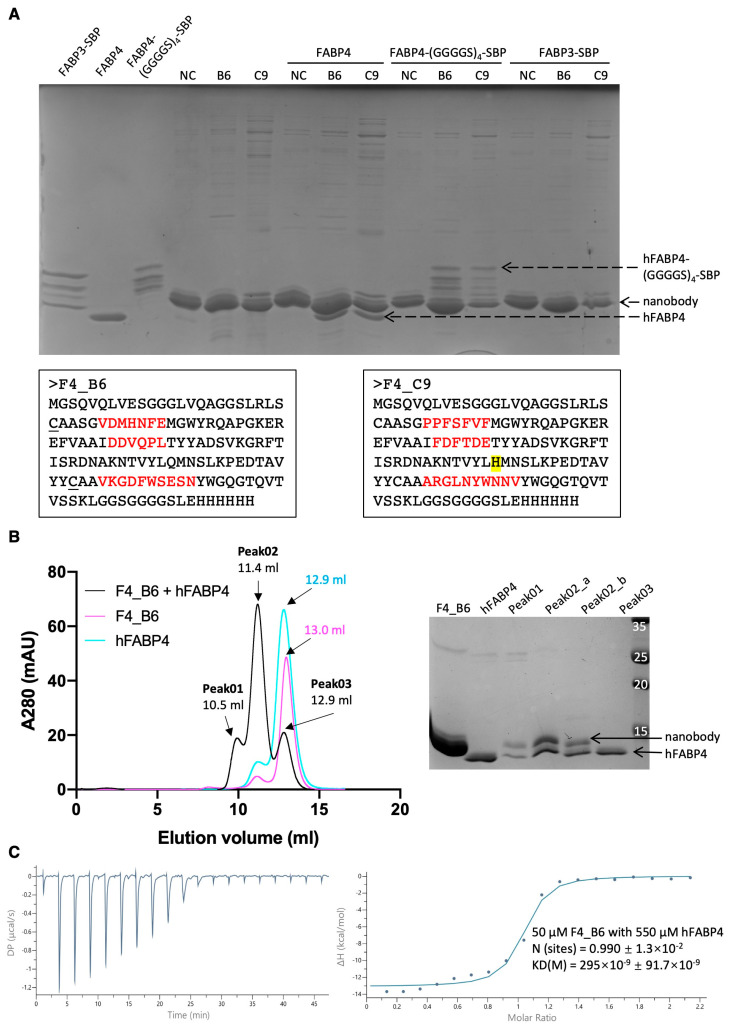
Analysis of the interaction between hFABP4 and hFABP4-specific nanobodies—F4_B6 (or B6) and F4_C9 (or C9). (**A**) Protein sequences of F4_B6 and F4_C9, and SDS-PAGE image of the pull-down assay, including the Ni-NTA beads coated with nanobodies, the elution fraction after incubation with hFABP4, hFABP4-(GGGGS)_4_-SBP, or FABP3-SBP. NC refers to the negative control, which is a nanobody that does not interact with hFABP4. (**B**) SEC profiles and SDS-PAGE image of F4_B6, hFABP, and the mixture of F4_B6 with hFABP4. (**C**) Analysis of the affinity between F4_B6 and hFABP4 via ITC. Raw ITC data (**left**) and fitted curves (**right**) are shown. 50 µM F4_B6 in cell with 550 µM hFABP4 in syringe.

**Figure 7 antibodies-14-00039-f007:**
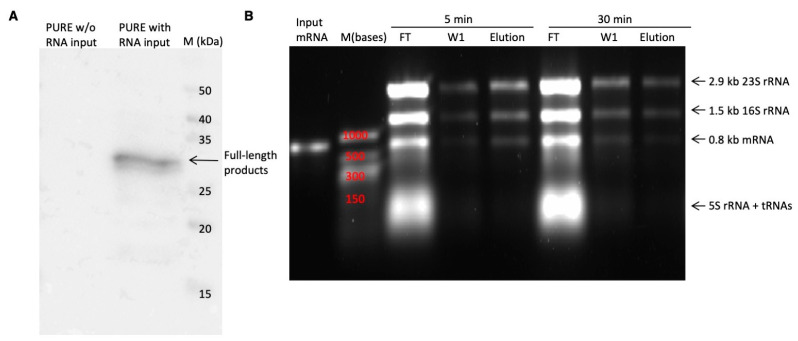
Optimization of PURE ribosome display by modifying spacer genes. (**A**) Western blot image of proteins synthesized by the PURE system. (**B**) RNA agarose gel image showing the efficiency of PURE ribosome display after using TolA as spacer gene under different incubation times.

## Data Availability

No new data were created.

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
