# Peer review of "Validation and Optimization of PURE Ribosome Display for Screening Synthetic Nanobody Libraries"

_2073-4468, 2025, doi:10.3390/antib14020039_

Round 1

Reviewer 1 Report

Comments and Suggestions for Authors

The authors generated a synthetic Nanobody library and tested the PURE in vitro transcription/translation system to select antigen binding fragments. While this approach should be (theoretically) an excellent tool to retrieve rapidly the target affinity reagents, only very few papers seem to be successful and most researchers switch back to the original immune Nanobody libraries and phage display selections. 

In this report the authors provide useful insights in the pitfalls that might be encountered with the synthetic libraries and selection using the PURE expression system. 

The weak points of this work is that the final affinity reagents are of relative low affinity (KD in the order of 100 nM), which is for therapeutic applications most likely insufficient, so that additional affinity maturation steps will be required. 

A second weak point  concerns the importance given (according to the authors) to the 3'end mRNA spacer (or 5-ending oligopeptide). (quote line 406 and 407 from the Conclusion:"In this work, we highlighted the importance of spacer gene choice in ribosome display when using the PURE system." However, in the results section they don't show (in figure or by quantification) a comparison of enrichment or efficiency in expression of the RMP. In contrast this was investigated and shown in their first chapters to evaluate or optimise the PURE tool with Nb2 (Figure 3 and Figure 4). This means that the reader has to accept this conclusion without seeing hard data. 

Author Response

Response to Comments

Reviewer #1:

  1. The weak points of this work is that the final affinity reagents are of relative low affinity (KD in the order of 100 nM), which is for therapeutic applications most likely insufficient, so that additional affinity maturation steps will be required. 

The Kd values of nanobodies selected from synthetic libraries were not as high as those from immune libraries, due to the absence of an in vivo maturation step. For therapeutic applications, additional affinity maturation steps would be necessary. However, for diagnostic applications like ELISA or Western blotting probes, moderate-affinity nanobodies can still be functional.

  1. A second weak point concerns the importance given (according to the authors) to the 3'end mRNA spacer (or 5-ending oligopeptide). … However, in the results section they don't show (in figure or by quantification) a comparison of enrichment or efficiency in expression of the RMP. In contrast this was investigated and shown in their first chapters to evaluate or optimise the PURE tool with Nb2 (Figure 3 and Figure 4). This means that the reader has to accept this conclusion without seeing hard data. 

We have added the efficiency in expression of functional RMP for the TolA spacer in the revised version (lines 359-362).

Reviewer 2 Report

Comments and Suggestions for Authors

This study addresses critical limitations in PURE ribosome display, a cell-free protein display technique, by systematically optimizing spacer gene selection, translation kinetics, and binder enrichment strategies. The authors construct a synthetic nanobody library with randomized CDRs and a high-frequency framework, employing trimer phosphoramidites to minimize synthesis errors. They successfully isolate EGFP- and hFABP4-specific nanobodies, demonstrating the system's robustness even with suboptimal spacer genes. Key innovations include real-time PCR monitoring of enrichment, protein pre-elution, and strategies to reduce non-specific binding. 

Although the following concerns need to be addressed by the authors.

  1. While the study highlights PURE system advantages, a direct comparison with cell-extract ribosome display (e.g., efficiency, yield) would strengthen its claims.
  2. Lines 402-403: «Following this validation and optimization, the application of PURE ribosome display for screening large libraries is expected to grow exponentially in the near future». Although the authors predict exponential growth in large-library screening, no data on library size or diversity limits are provided.
  3. There are some minor typos in the manuscript.
  4. Please check the reported references also in accordance with the format required by “Antibodies”
Comments on the Quality of English Language

The English could be improved to more clearly express the research.

Author Response

Response to Comments

Reviewer #2:

  1. While the study highlights PURE system advantages, a direct comparison with cell-extract ribosome display (e.g., efficiency, yield) would strengthen its claims.

The reported efficiency of S30-extract ribosome display for a short His6-tag peptide is approximately 5%, which is comparable to the performance of the PURE system when using a suboptimal spacer gene (M13 pIII) for larger proteins like nanobodies. This comparison has been incorporated into the revised main text (lines 182-184).

  1. Lines 402-403: «Following this validation and optimization, the application of PURE ribosome display for screening large libraries is expected to grow exponentially in the near future». Although the authors predict exponential growth in large-library screening, no data on library size or diversity limits are provided.

Indeed, we have no data on the diversity limit. To soften the statement, we have deleted the word “exponentially”.

  1. There are some minor typos in the manuscript.

Typos have been corrected.

  1. Please check the reported references also in accordance with the format required by “Antibodies”

References have been reformatted.

  1. The English could be improved to more clearly express the research.

We have revised the manuscript thoroughly (please refer to the marked version in pdf).

Round 2

Reviewer 2 Report

Comments and Suggestions for Authors

I recommend accepting manuscript in present form